# Presence and Persistence of Andes Virus RNA in Human Semen

**DOI:** 10.3390/v15112266

**Published:** 2023-11-17

**Authors:** Roland Züst, Rahel Ackermann-Gäumann, Nicole Liechti, Denise Siegrist, Sarah Ryter, Jasmine Portmann, Nicole Lenz, Christian Beuret, Roger Koller, Cornelia Staehelin, Andrea B. Kuenzli, Jonas Marschall, Sylvia Rothenberger, Olivier Engler

**Affiliations:** 1Spiez Laboratory, Swiss Federal Office for Civil Protection, 3700 Spiez, Switzerlandolivier.engler@babs.admin.ch (O.E.); 2ADMED Microbiology, 2300 La Chaux-de-Fonds, Switzerland; 3Food Microbial Systems, Risk Assessment and Mitigation Group, Agroscope, 3097 Bern, Switzerland; 4Institute for Infectious Diseases, University of Bern, 3001 Bern, Switzerland; 5Department of Infectious Diseases, Inselspital, Bern University Hospital, University of Bern, 3010 Bern, Switzerland; 6Institute of Microbiology, University Hospital Center and University of Lausanne, 1005 Lausanne, Switzerland

**Keywords:** Andes virus, persistence, semen, neutralizing antibodies

## Abstract

When infecting humans, Andes orthohantavirus (ANDV) may cause a severe disease called hantavirus cardiopulmonary syndrome (HCPS). Following non-specific symptoms, the infection may progress to a syndrome of hemorrhagic fever combined with hyper-acute cardiopulmonary failure. The case fatality rate ranges between 25–40%, depending on the outbreak. In this study, we present the follow-up of a male patient who recovered from HCPS six years ago. We demonstrate that the ANDV genome persists within the reproductive tract for at least 71 months. Genome sequence analysis early and late after infection reveals a low number of mutations (two single nucleotide variants and one deletion), suggesting limited replication activity. We can exclude the integration of the viral genome into the host genome, since the treatment of the specimen with RNAse led to a loss of signal. We demonstrate a long-lasting, strong neutralizing antibody response using pseudovirions expressing the ANDV glycoprotein. Taken together, our results show that ANDV has the potential for sexual transmission.

## 1. Introduction

Orthohantaviruses are enveloped, spherical viruses belonging to the family Hantaviridae, order Bunyavirales [1]. Their single-stranded negative-sense genome comprises three segments, S (small), M (medium) and L (large), encoding the nucleoprotein (N), envelope proteins (Gn and Gc), and the L protein or viral RNA-dependent RNA polymerase, respectively [2]. At present, over 28 hantavirus species are known to be pathogenic in humans. Each virus is usually closely associated with a single rodent (or insectivore) species as the reservoir host. As a consequence, the geographical distribution of the different Orthohantaviruses is determined by the presence of the respective hosts [3,4,5,6]. Orthohantaviruses are grouped into the so-called “Old World” viruses in Europe and Asia, and “New World” viruses in North and South America. Typically, two different hantavirus-caused clinical syndromes have been reported: hemorrhagic fever with renal syndrome (HFRS) caused by Old World hantaviruses and hantavirus cardiopulmonary syndrome (HCPS) caused by New World hantaviruses [3].

Andes orthohantavirus (ANDV) is endemic in Chile and Argentina [7,8]. Five different lineages of ANDV have been proposed (Central Plata, Central Buenos Aires, Central Lechiguanas, North, and South) [9]. The primary reservoir of ANDV is the long-tailed pygmy rice rat (Oligoryzomys longicaudatus) [8]. Transmission to humans occurs primarily via the inhalation of virus-containing aerosols from rodent excretions such as urine, feces and saliva [3,10]. As an exception among hantaviruses, person-to-person transmission has been demonstrated for ANDV, with virus transmission likely taking place during the prodromal disease phase or shortly after. Being a sex partner of, or sleeping in the same room as, a patient and exposure to body fluids from a patient are risk factors for secondary infections [11,12].

The incubation period for ANDV infections is seven to 39 days (median 18 days) [13]. The clinical course of HCPS generally progresses through a prodromal, a cardiopulmonary, and a convalescent phase. The prodromal phase is characterized by nonspecific symptoms such as a high fever, chills, myalgia, nausea, headache, vomiting, abdominal pain, and diarrhea, lasting for three to six days. In about 50% of infections, this is followed by a swift progression to the cardiopulmonary phase, which is characterized by dyspnea, cough, tachycardia, and hypotension as a consequence of rapidly progressing pulmonary edema due to capillary leakage. Hemorrhagic manifestations in various sites (hematuria, intestinal bleeding, metrorrhagia) are observed as a consequence of thrombocytopenia and disseminated intravascular coagulation, and indicate poor prognosis. Cardiogenic shock is the main cause of death, which may occur within hours of the cardiopulmonary phase. Case fatality rates range from about 20 to 40% [14]. Patients surviving the acute stage enter the convalescent polyuric stage, during which the pulmonary edema is resolved. The treatment of HCPS is primarily supportive [3,11].

The diagnosis of HCPS is based on clinical symptoms, epidemiological data, and laboratory tests [3]. During the prodromal phase, diagnosis is hampered by the lack of specific symptoms or diagnostic findings. In contrast, during the cardiopulmonary phase, a presumptive diagnosis may be established based on exposure history, the presence of pulmonary edema and certain hematological parameters such as thrombocytopenia, hemoconcentration, a pronounced left-shift of granulocytes and the presence of lymphocytes with immunoblastic morphological features (>10%) [15]. The definitive diagnosis of HCPS is usually based on the serological testing of IgM and IgG antibodies, which appear during the late prodromal or the beginning of the cardiopulmonary phase. In addition, viral RNA has been detected via reverse transcription polymerase chain reaction (RT-PCR) in peripheral blood cells or serum up to 15 days before and 35 days after the onset of symptoms [16], and may be found in the urine, gingival crevicular fluid, saliva, semen, and respiratory samples of patients [17,18,19]. 

In 2016, we reported on two imported, severe but non-fatal cases of HCPS due to ANDV infection in Switzerland [19]. In this study, we present a six-year follow-up of the detection of viral RNA in semen samples from the male patient and characterize the neutralizing antibody (nAbs) titers in serum using a pseudovirion neutralization test. We provide whole-genome analyses of the virus present in semen samples early after the onset of symptoms and six years after the onset of symptoms. Virus isolation was attempted but not successful for any of the samples or culture systems used.

## 2. Materials and Methods

### 2.1. Patient and Case Report

In this study, we describe the long-term post-infection follow-up of a 55-year-old man. He had trekked from Ecuador to Chile from September to November 2016 and became ill upon his return to Switzerland in December 2016. He was diagnosed with HCPS caused by ANDV, which was confirmed via RT-qPCR as well as via positive immunoglobulin (Ig) M and IgG serology. A detailed case report on the acute disease phase has been published previously [19]. 

### 2.2. Samples

The long-term monitoring of viral load, determination of antibody titers, and viral sequence analyses were performed in blood and/or semen samples during his illness and up to 71 months post infection. Blood samples were collected at indicated time points in tubes containing EDTA as anticoagulant (Vacutainer^®^, Becton Dickinson and Company, Allschwil, Switzerland), and plasma was obtained by centrifuging the tubes for 20 min at 1500× *g*. Semen samples were collected at indicated time points in sterile containers. As negative controls, five semen samples from healthy donors were tested. 

### 2.3. Monitoring of Viral Load by RT-qPCR

RNA was extracted using the EZ1 Advanced XL instrument using the EZ1&2 Virus Mini Kit v2.0 (Qiagen, Hombrechtikon, Switzerland) and analyzed using RT-qPCR. The viral load in semen samples was monitored via ANDV-specific RT-qPCR using two independent targets on segment S with the following primers and probes: forward 1: 5′-GCAGCTGTGTCTACATTGGAGAC-3′, reverse 1: 5′-GCTCCTATAGCCTTCCAATCAGC -3′, probe 1: 5′-FAM-ACAAAACCAGTTGATCCA-3′ and forward 2: 5′-GAATGAGCACCCTCCAAGAATTG-3′, reverse 2: 5′-CGAGCAGTCACGAGCTGTTG-3′ probe 2: 5′-FAM-ACATCACAGCACACGA-3′. Where plotted, genome equivalents for viral quantification using RT-qPCR were calculated using a standard curve based on cycle threshold values from serially diluted RNA and a linear regression model [19]. 

### 2.4. Separation of Cells from Seminal Plasma by Centrifugation

One milliliter of ejaculate was centrifuged for 15 min at 600× *g*. The supernatant was carefully transferred into a new tube and the pellet resuspended in 1 ml of PBS. Then, 100 μL of initial input, supernatant, and resuspended pellet were inactivated in AVL (Qiagen, Switzerland) and EtOH at a ratio of 1:4:4. RNA was extracted using the EZ1 Advanced XL instrument using the EZ1&2 Virus Mini Kit v2.0 (Qiagen, Switzerland) and analyzed via RT-qPCR as described above.

### 2.5. Determination of Viral Nucleic Acid Type

To determine whether the obtained ANDV-specific real-time PCR signal is derived from the viral genome possibly integrated into the host DNA, we isolated nucleic acid from the ejaculate using the EZ1 Advanced XL instrument and treated it with RNase A (Thermo Fisher Scientific, Basel, Switzerland) according to the manufacturer’s protocol. As controls, RNA derived from a Sin Nombre Virus (SNV) culture and a plasmid containing the sequence targeted by the SNV-specific real-time PCR were used. For the ANDV-specific real-time PCR, primers were used, as described above. For the SNV-specific real-time PCR, the following primers were used: forward: 5′-TGGACCCCGATGATGTTAACA-3′, reverse: 5′-CCARTTTCTGAGCTGCAATAAGATC-3′, probe: 5′-FAM-ACGGGCAGCTGTGTCTGCATTGG-3′.

### 2.6. Pseudovirion Neutralization Assay (PsVNA)

The PsVNA was performed as described elsewhere [20] using vesicular stomatitis virus (VSV) pseudotyped with ANDV glycoproteins as a substitute for infectious ANDV. The expression plasmid pI18 for the glycoprotein of ANDV strain CHI-7913 was kindly provided by Nicole Tischler (Molecular Virology Laboratory, Fundación Ciencia & Vida, Santiago, Chile) and has been described previously [21].

### 2.7. Sequencing

RNA was isolated from 100 µL of human semen using the RNeasy Plus Universal kit (Qiagen, Switzerland) according to the manufacturer’s instructions, followed by reverse transcription using Superscript IV (Thermo Fisher Scientific, Basel, Switzerland). For targeted amplification, primers were designed using Primal 19 for all three segments using the following reference sequences: KY659432.1, KY604962.1 and KY659431.1. Primer sequences and volumes are available upon request. cDNA was amplified using multiplex PCR as described previously [22] using 35 amplification cycles. PCR fragments were pooled and sequenced on a GridION using the native barcoding kit (EXP-NBD104) in combination with the genomic sequencing kit (SQK-LSK109). Consensus sequences of the individual segments were created using the artic pipeline [23] adapted to our ANDV primer panel. Sequences have been uploaded to NCBI (OR405520–OR405525). Variants between different time points were identified using snippy v.4.6.0 [24].

### 2.8. Phylogenetic Analysis of ANDV

For phylogenetic analysis, nucleotide sequences of each segment of previously sequenced Andes orthohantavirus were retrieved from NCBI and aligned using MAFFT v.7.505 [25]. A phylogenetic tree was constructed using IQ-Tree v.2.2.0.3 [26], including automatic model selection and 1000 ultrafast bootstrap to obtain branch supports [27]. The trees of each segment were rooted using SNV sequences (L25784, L25782, L37901) as outgroup and visualized using ITOL [28].

### 2.9. Virus Isolation on Cell Culture

Virus propagation was attempted using three different semen samples collected 40, 82, and 320 days after the onset of symptoms, respectively. These samples were selected for their low CT values. Fresh or frozen ejaculate was added either to Vero E6 cells, a 1:1 mix of BSR/Vero cells, primary human epithelial cells from bronchial (hAECB) or nasal (hAECN) biopsies, or three-dimensional human airway epithelia (MucilAir^TM^), respectively. Four (MucilAir^TM^) or ten days (Vero E6, BSR/Vero, hAECB, and hAECN) post infection, the cell suspension was passaged onto fresh cells. This step was repeated three (BSR/Vero, hAECB, and hAECN), six (Vero E6) or seven (MucilAir^TM^) times, respectively. In addition to the use of original sample material, homogenized sperm cells were used for infection. Briefly, the semen sample was mixed 1:1 with PBS and centrifuged. The supernatant was discarded, and the pellet was washed with PBS. Cells were pelleted again via a centrifugation step, followed by washing with PBS. The pellet was finally resuspended in PBS and homogenized. Further details and the respective protocols are described in Appendix A.

## 3. Results

### 3.1. Long-Term Monitoring of Viral Load by RT-qPCR

During the acute phase of disease, viral RNA was detectable in the blood, urine, respiratory and semen samples. Viral RNA was undetectable in the urine, respiratory samples, and blood samples 15, 54, and 172 days after the onset of symptoms [19]. RNA remains detectable until the present time (2188 days post infection) in semen samples (Figure 1).

### 3.2. Analysis and Allocation of Viral Material

While untreated ejaculate had a Ct-value of approximately 26.7, the supernatant cleared of cells via centrifugation gave a Ct-value of 30. This equals a 90% loss of signal. Inversely, the cell pellet resuspended in equal volume led to a signal of Ct 27, demonstrating only a 20% loss of signal (Figure 2). We thus conclude that the virus is mostly located intracellularly in one of the cell types found in semen. To further determine if the viral genome had integrated into the host DNA, we performed RNAse digestion of the test material. Upon digestion with RNAse, the RT-qPCR signal was lost (Figure 3).

### 3.3. Neutralizing Antibody Titers (nAbs)

At the time of admission (day 1), the patient displayed a robust total IgG titer with detectable viremia. Although the virus-neutralizing capacity was modest at day 1, the patient quickly developed a very strong nAbs response, reaching a titer of over 30,000 on day 20 in our assay. The neutralizing capacity decreased but stabilized at a high titer during the observation (Figure 4). The last time point was measured in a separate run, using a different instrument. As a negative control, samples from a naïve donor were included in each run, ranging from one to five in magnitude.

### 3.4. Viral Whole Genome Sequence Analyses

Sequencing was performed on semen samples collected 247 days and 1978 days after the onset of symptoms. Target amplification resulted in an almost complete genome coverage of all three segments at both time points (>94% for segment S, >98% for segment M and >90% for segment L). Comparison of the early and late isolates revealed a 33 bp deletion in the non-coding region of the S segment (position 1451), as well as two single nucleotide variants within the M segment (G815A: K265E, A2250G: R743Q) and one single nucleotide variant within the L segment (A3555G: K1119R). Coordinates are given relative to ANDV-CH-LS-2016. Phylogenetic analysis shows a close relationship to a previously sequenced ANDV from Argentina and Chile (Figure 5).

## 4. Discussion

We present the long-term follow up of virus persistence in the semen samples of a patient after full recovery from ANDV HCPS. We detected high neutralizing titers in the blood 6 years after the onset of the disease, implying a long-lasting immune response. This finding is in line with previous studies showing the presence of nAbs years after Puumala Virus (PUUV), SNV and ANDV infections [29,30]. 

High nAbs on hospital admission have been previously correlated with less severe HCPS [31]. More recently, a similar observation was made by analyzing neutralizing antibody titers in three cases of Puumala virus infection in a family in Switzerland [32]. These observations suggest that the rapid development of high levels of neutralization antibodies contribute to survival. Several studies have shown that the humoral response towards hantaviruses is long lasting, and that neutralizing antibodies persist at high levels for years in the sera of convalescent patients [29,30,33,34].

Repeated symptomatic infection with hantaviruses have not been observed, suggesting life-long protection [35]. Our finding of very high nAb titers does not seem to be an exceptional case, as it has been reported previously [36]. These findings are compatible with a possible re-stimulation of antiviral B-cell responses due to viral persistence. Further studies addressing the persistence of ANDV will be needed to confirm this hypothesis.

Little is known about the kinetics of appearance of nAbs during the acute phase of the disease and the evolution of nAb titers during the convalescent phase of the disease. Here, we show that the patient developed a strong immune response, with nAb titers reaching peak levels at day 21 after hospital admission. The strong increase in nAb titers three weeks after patient hospitalization highlights the importance of measuring the dynamics of the antibody response over several weeks to assess the correlation between nAb dynamics and disease severity. Our study indicates that the nAb titer remains stable over time. Although analysis of more samples would be needed to confirm this finding, an increase in neutralizing antibodies months to years after the acute infection has been previously documented, suggesting that latent antigenic stimulation may be possible [36]. 

Viral RNA remained primarily detectable intracellularly in semen samples throughout the complete study period of almost six years. The virus remained persistent, without integration into the genome of the host, as demonstrated by the loss of signal after the RNAse treatment of the analyzed samples. Viral whole genome sequence analyses of samples from the beginning of disease and six years later revealed a 33 bp deletion in the genomic segment S. As RNA viruses show extremely high genetic variability, with substitution rates ranging from 10^−6^ to 10^−4^ substitutions per nucleotide per cell infection [37], this result suggests that ANDV persisted within cells of the male reproductive tract with only very limited replication activity. So far, ANDV is the only hantavirus for which person-to-person transmission has been documented [12,16,38,39,40,41]. However, the precise routes of transmission have not been identified. The inhalation of droplets or aerosolized virions are the most likely routes of infection [42]. Although transmission via sexual contact has not been documented, high viral loads and extensive contact among people may contribute to a higher likelihood of transmission [42]. Viral mutations and single-nucleotide polymorphisms are additional factors that could influence ANDV virulence and/or transmissibility [43]. 

It is known that viruses may persist and slowly spread from cell to cell without the production of an infectious virus [44,45,46]. To establish persistent infections, viruses must avoid elimination by the immune response of the host; this may be achieved by establishing infections in immunologically privileged sites such as the testes. Additionally, whilst maintaining their genomes within cells, viruses must avoid infected cells being killed [47]. Interestingly, six different Orthohantaviruses, including ANDV, have been shown to inhibit apoptosis [48], thereby preventing the elimination of virus-infected cells.

Unfortunately, the isolation of the infectious virus was not successful for any of the samples or culture systems used (Appendix A). A successful isolation of ANDV using Vero E6 cells from a sample taken from a 10-year-old Chilean boy two days before he became febrile has been described [49]. However, it cannot be ruled out that the cell lines used are not susceptible to the ANDV strain from this study. Given the high susceptibility and lethality of ANDV in Syrian hamster [50], an isolation from this host might be successful. Unfortunately, we do not have an animal facility. Several approaches using serial passages in animal models have efficiently been applied to other Orthohantaviruses [51,52] and might also be used for ANDV. Unsuccessful isolation, however, is not an indication of the absence of infectious particles, since Orthohantaviruses are notoriously difficult to isolate [53]. 

After separating cells from the seminal plasma, viral RNA was detectable mostly in the cells and, to a lesser extent, in the seminal plasma fraction, indicating the cellular association of the virus. It remains to be determined which cell type carries the virus. Besides spermatozoal cells, ejaculates contain germinal elements, neutrophils, macrophages, lymphocytes, epithelial cells and Sertoli cells [54]. Since the testes are immunologically privileged sites, the virus may persist in this organ in the absence of active replication. Many factors may influence persistence, including virus-associated factors, such as the level of viraemia, virus structural stability, and the ability of the virus to replicate within the male reproductive tract. Mechanisms of immune evasion and host factors may also contribute to persistence, including inflammatory mediators altering the blood barrier permeability, systemic immunosuppression, the immune response of the male reproductive tract, or the presence of sexually transmitted diseases [55]. There is evidence that at least 27 different viruses, across a broad range of virus families or orders, can be found in human semen. While many of these viruses cause chronic or latent infections, some cause acute infections, including Zika virus, Ebola virus, Lassa mammarenavirus, Chikungunya virus, and Rift Valley fever phlebovirus [55]. To our knowledge, this is the first report of a virus of the order Bunyavirales persisting for almost six years in the semen samples of a patient. The limitation of this study is the small sample size. It remains to be determined whether persistence occurs in a larger population of long-term Andes virus disease survivors.

## Figures and Tables

**Figure 1 viruses-15-02266-f001:**
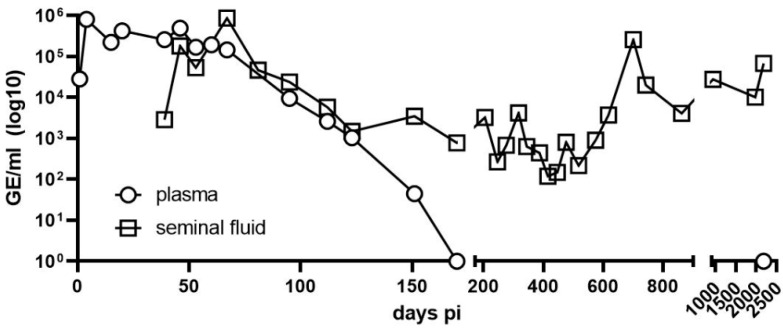
Detection of Andes virus RNA in plasma and semen. RNA levels from plasma (circles) and semen (squares) were assessed on the indicated days post infection using in-house quantitative real-time polymerase chain reaction; viral load on day 1 was from a serum sample, and from day 4 onward was from EDTA blood.

**Figure 2 viruses-15-02266-f002:**
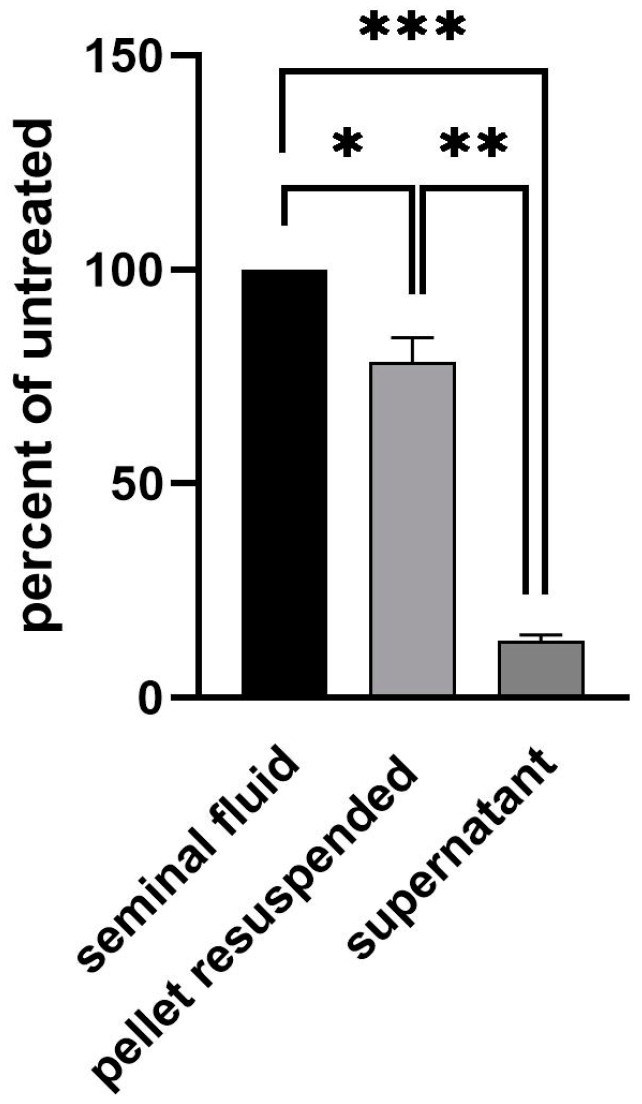
Location of viral material in semen. In total, 1 ml of semen was centrifuged at 600× *g* for 15 min. The supernatant was transferred into a new tube and the pellet resuspended in 1 ml of PBS. Nucleic acid from 100 μL of input semen, supernatant and resuspended pellet was isolated and analyzed using real-time PCR. Statistical analysis was performed using unpaired the Student’s t-test (***, *p* < 0.001; **, *p* < 0.01; *, *p* < 0.05).

**Figure 3 viruses-15-02266-f003:**
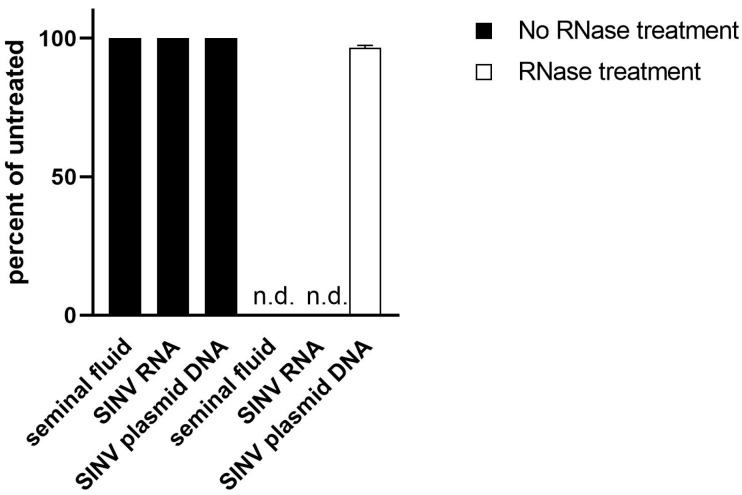
Analysis of viral material. Nucleic acid from semen was isolated and left untreated (black) or subjected to RNase treatment (white). SNV RNA and a plasmid (not sensitive to RNase treatment) containing the target of the SNV RT-qPCR served as controls. After RNase digestion, no signal could be detected in material derived from semen or control RNA (n.d.; not detected).

**Figure 4 viruses-15-02266-f004:**
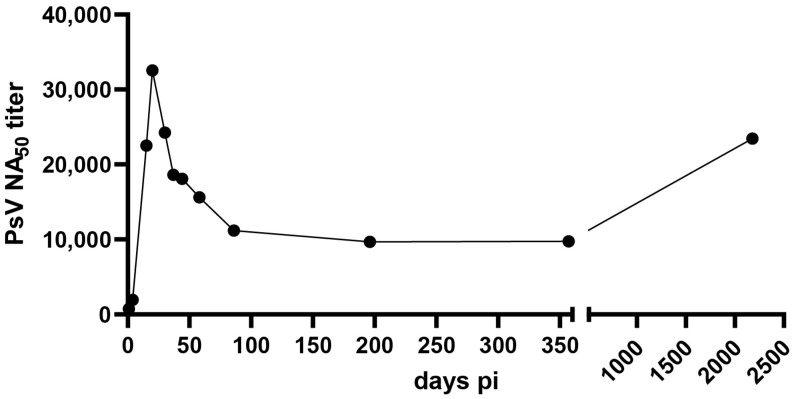
Neutralization of ANDV using patient sera. The neutralizing antibody titer (reciprocal IC_50_) against ANDV in serum samples was assessed using a pseudovirion assay. Half maximal inhibitory concentrations were estimated via a model of nonlinear regression fit with settings for log (inhibitor) vs. normalized response curves using GraphPad Prism v9.

**Figure 5 viruses-15-02266-f005:**
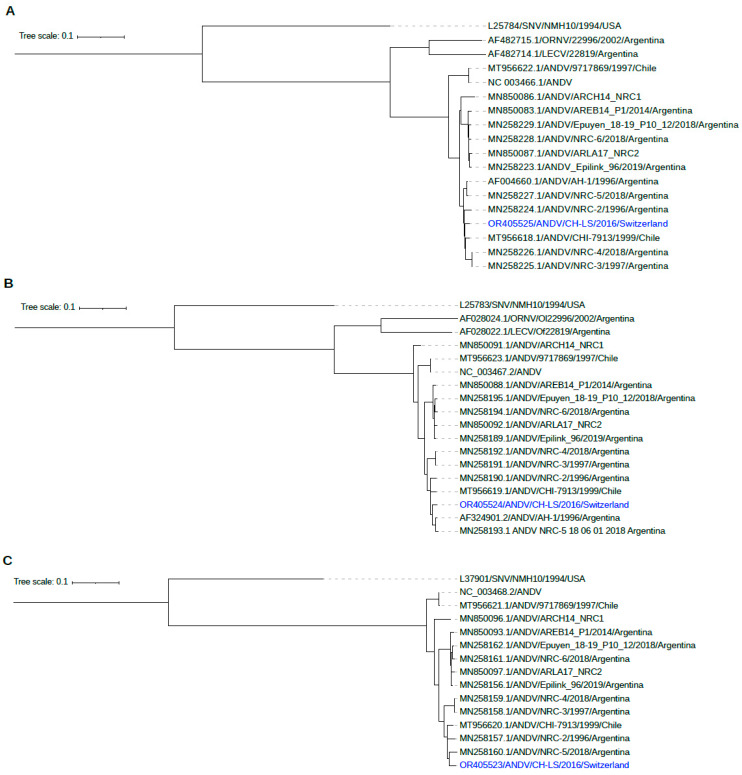
Phylogenetic analysis of ANDV shows a close relationship to previously sequenced ANDV from Chile and Argentina ((**A**): segment S, (**B**): segment M, (**C**): segment L).

## Data Availability

Sequences have been uploaded to NCBI (OR405520–OR405525).

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
