# Peer review of "Presence and Persistence of Andes Virus RNA in Human Semen"

_viruses, 2023, doi:10.3390/v15112266_

Round 1

Reviewer 1 Report

Comments and Suggestions for Authors

The paper "Presence and long-lasting persistence of Andes Virus RNA in human semen" is an interesting read and this reviewer deems it acceptable for publication with the following revisions in place:

Major Points:

Lines 172-176: The claim on lines 175-176 that the virus is found intracellularly depends upon the comparison of signal between the supernatant and the pellet. This comparison should be described and a test for significance between these two groups should be shown on figure 2.

Line 175-176: The sentence spanning these lines claims with absolute language that the virus is located intracellularly, yet there is still detection of virus in the supernatant. Either this claim should be adjusted to say that most of the virus is found intracellularly, or a value should be established at which point detection should be understood by the reader as negligible.

Figure 2: As stated previously, significant differences between the ‘pellet suspended’ and ‘supernatant’ groups should also be shown. Significant differences critical to the conclusions should be noted in the figure description.

Line 242-243: The first sentence of this paragraph states that viral RNA were detectable intracellularly in semen samples throughout the course of the study, implying that that they were not detectable extracellularly. This statement should be balanced either with a qualifier such as “primarily intracellularly,” or there should be justification given previously (see comment about lines 175-176) about why the detection seen in the supernatant should be considered negligible.

Supporting Information, lines 9-12: There is mention of patient 2 - is this the same patient 2 from reference [15]? If two patients were indeed sampled, this is not made clear in the manuscript and should be noted, along with accompanying figure data. 

Minor Points:

Title: The word "long-lasting" is redundant. Consider removing.

Title: The letter "v" in word "Virus" should not be capitalized.

In several instances, a number is used to start a sentence, which is generally considered unacceptable. Consider changing the numbers (e.g. from "10 ml" to "Ten milliliters") to words where this occurs.

Line 36: Add ", respectively" after the word "polymerase".

Line 36: Change "pathogenic for" to "pathogenic to" or "pathogenic in".

Line 54: Please cite the original reference for this information rather than [4].

Line 88: Missing period at the end of sentence.

Lines 90-94: It should be explicitly stated in this paragraph that the patient was diagnosed with HCPS caused by Andes virus.

Line 123: Add the word "the" before "EZ1 Advanced...".

Line 133: Typo in “… ANDV glycoproteins as s a substitute …”

Line 163: Do the respiratory samples mentioned here refer to the tracheobronchial secretion samples described in Supporting Information line 12? If not, please describe what "respiratory samples" are.

Line164: Add ", respectively (data not shown)" after the word "symptoms" or add these data to figure 1. Adding the data to figure 1 would be preferable to this reviewer.

Figure 3: Define "(n.d.)" and change "SNV DNA" to "SNV plasmid DNA" for clarification.

Line 194: Define "nAbs" and change "30'000" to "30,000".

Line 202: Change "IC50" to "IC50".

Line 207: Change "performed of semen samples" to "performed on semen samples" and change "from day 247 and day 1978" to "collected 247 days and 1978 days after the onset of symptoms". 

Line 250-251: Please provide a reference for the statement that viruses can slowly spread without producing infectious virus.

Line 261-262: As noted in the sentence that follows, Orthohantaviruses are difficult to isolate, and therefore the claim on lines 261-262 that the viraemic phase may precede symptoms is not made on a directly applicable basis.

Lines 265-266: See comment about lines 175-176.

Supporting Information, lines 34, 39, 46: Isn't DMEM (not MEM) generally used for Vero E6 cells?

Supporting Information, lines 41, 63, 65, 83, 84: Change "37°" to "37°C". 

Supporting Information, lines 41, 63: Why was CO2 not used?

Supporting Information, lines 55, 60, 61, 64: Isn't MEM (not DMEM) generally used for Vero cells?

Supplementary Information, lines 66-68: Hantaviruses are often found in the supernatant of infected cells. Why did the authors not use cell supernatant or scraped cells resuspended in cell supernatant for subsequent infections?

Supplementary Information, line 73: Use a new paragraph after "passages."

Supplementary Information, line 90: Change "2'000" to "2,000".

Supplementary Information, line 118: Change: 4'100" to "4,100".

Author Response

Dear reviewer. Thanks a lot for the valuable comments. I highly appreciate the effort. Please find my comments in the attachment.

Reviewer 2 Report

Comments and Suggestions for Authors

Lines 159-159. Detection, or not, of infectious and potentially transmissible virus would be a key finding. It would be helpful to at least state the cells used for virus detection and the number of serial passages made and not make the hasty reader have to go to the supplementary protocol to find out. 

Lines 267-264. Persistence of virus infection in the male reproductive tract is of potential epidemiological significance if infectious virus is shed with the risk of sexual transmission. It would be of interest to the reader if the authors would comment on that risk even though they did not detect infectious virus on cell culture. Given that hantavirus isolation in cell cultures is difficult, a comment on the relative susceptibility of the cell cultures used would provide a helpful context for the negative results found. Is there a risk of false negatives? Are there any documented instances of sexual transmission of Andes virus, or just the possibility given that close physical contact is associated with person-to-person virus transmission? 

Author Response

Dear reviewer. Thanks a lot for the valuable comments. I highly appreciate the effort. Please find my comments in the attachment.:

Reviewer 3 Report

Comments and Suggestions for Authors

The paper is both novel, plausible and valuable as a long-term follow-up to the authors initial 2018 paper which showed that hantaviruses join the group of viruses present long-term in semen along with bunyavirus RVF and other viruses. The authors expand on their initial report by describing possible adaption mutations in the virus and nAb titers over a long time period. 

The authors should discuss the extremely prolonged positivity for viral RNA in serum in relationship to semen expression and nAb expression - it is more likely that the findings in this exceptional case are not generalisable to acute lethal infection, however it is possible they contributed to survival.

Introduction is concise and mostly accurate, however the case fatality rate of 25-40% (line 66) is questionable and should be validated either by references or by modifying it to "of around 20-40%" or similar; note Alonso et al 2019.

Author Response

(The authors gave the same response as above.)
